# Study of Low-Frequency Hydroacoustic Waves' Behavior at the Shelf of Decreasing Depth

**Grigory I. Dolgikh [1],\*, Shengchun Piao [2], Sergey S. Budrin [1], Yang Song [2], Stanislav G. Dolgikh [1], Vladimir A. Chupin [1] , Sergey V. Yakovenko [1], Yang Dong [2] and Xiaohan Wang [2]**

[1] Il'ichev Pacific Oceanological Institute, Far East Branch of the Russian Academy of Sciences, Vladivostok 690041, Russia; ss_budrin@mail.ru (S.S.B.); sdolgikh@poi.dvo.ru (S.G.D.); chupin@poi.dvo.ru (V.A.C.); ser_mail@poi.dvo.ru (S.V.Y.)

[2] Acoustic Science and Technology Laboratory, Harbin Engineering University, Harbin 150001, China; piaoshengchun@hrbeu.edu.cn (S.P.); song_yang@hrbeu.edu.cn (Y.S.); dong_yang@hrbeu.edu.cn (Y.D.); wangxiaohan@hrbeu.edu.cn (X.W.)

\* Correspondence: dolgikh@poi.dvo.ru; Tel.: +74232312352

**Abstract:** In research into various hydrophysical and hydroacoustic wave processes, it is extremely important to know the regularities of their propagation in the sea at decreasing depths, especially in the shelf areas, and also to know the regularities of their transformation into seismoacoustic processes in the earth crust. In the course of the processing and analysis of the experimental data of our complex experiment, in this paper we investigate these regularities. In our experiment, we used a low-frequency hydroacoustic transmitter that generated harmonic oscillations at the frequency of 22 Hz and received hydroacoustic systems with a shore laser strainmeter. It was established that hydroacoustic waves, propagating at the shelf of decreasing depth, transform into seismoacoustic waves at the depth of the sea equal to or less than a half-length of the hydroacoustic wave. A comparison of the results of this work with earlier-obtained results allows us to state that such regularities should be inherent to all hydrophysical and hydroacoustic processes.

**Keywords:** hydroacoustic waves; hydroacoustic transmitter; laser strainmeter; model; transformation; seismoacoustic waves

## 1. Introduction

In solving various applied problems associated with the study of natural and artificial processes and phenomena occurring in the World Ocean, hydroacoustic methods are widely used. They are focused not only on the study of the main characteristics of the researched objects but also on their identification and direction finding. One of these problems is the contactless remote study of the structure and composition of the marine earth crust, especially for areas covered with thick ice. This study is based on the use of low-frequency hydroacoustic transmitters and coastal receiving systems for seismoacoustic signals [1,2]. The use of coastal receiving systems for seismoacoustic signals, generated as a result of the transformation of hydroacoustic signals at the "water-bottom" boundary, allows us to study the structure and composition of the marine earth crust without the large-scale destruction of the ice cover, which makes the use of these methods more economical in comparison with traditional ones. In the development and application of these methods, it is important to reveal the regularities of the propagation of low-frequency hydroacoustic signals at the shelf of decreasing depth and their transformation into seismoacoustic signals that further propagate in the earth crust. Knowledge of these regularities is very relevant in the study of signals generated by natural and artificial marine objects, especially in low-frequency sound and infrasound ranges and when propagating from "deep"

to "shallow" seas. The same knowledge is necessary when studying the communication signals of sea animals and the peculiarities of their communication in the shelf zones. Based on the obtained knowledge, the inverse problem can consequently be solved—the development and creation of transmitting systems in specific frequency ranges oriented at the near-bottom propagation of the signals they generate, which is extremely important when establishing ultra-long-distance communication with bottom observatories and mining complexes. The first similar work carried out with a low-frequency hydroacoustic transmitter, generating signals at a frequency of 33 Hz, made it possible to establish some regularities for this frequency range [3]. However, the question remains—how do the regularities identified in [3] transform when the frequency of the transmitted signal is reduced? In this paper, we study some peculiarities of the propagation and transformation of hydroacoustic signals at the frequency of 22 Hz at the shelf of decreasing depth.

## 2. Description of the Experiment

The experiment was conducted in Vityaz' Bay of the Sea of Japan. The map of the experiment is shown in Figure 1. The research vessel was anchored at the "radiation point" (coordinates N 42°35.6827′, E 131°09.8707′) at a depth of 32 m. From its board, a low-frequency hydroacoustic transmitter of the electromagnetic type was immersed to a depth of 18 m [4], generating in the water harmonic signals at a frequency of 22 Hz.

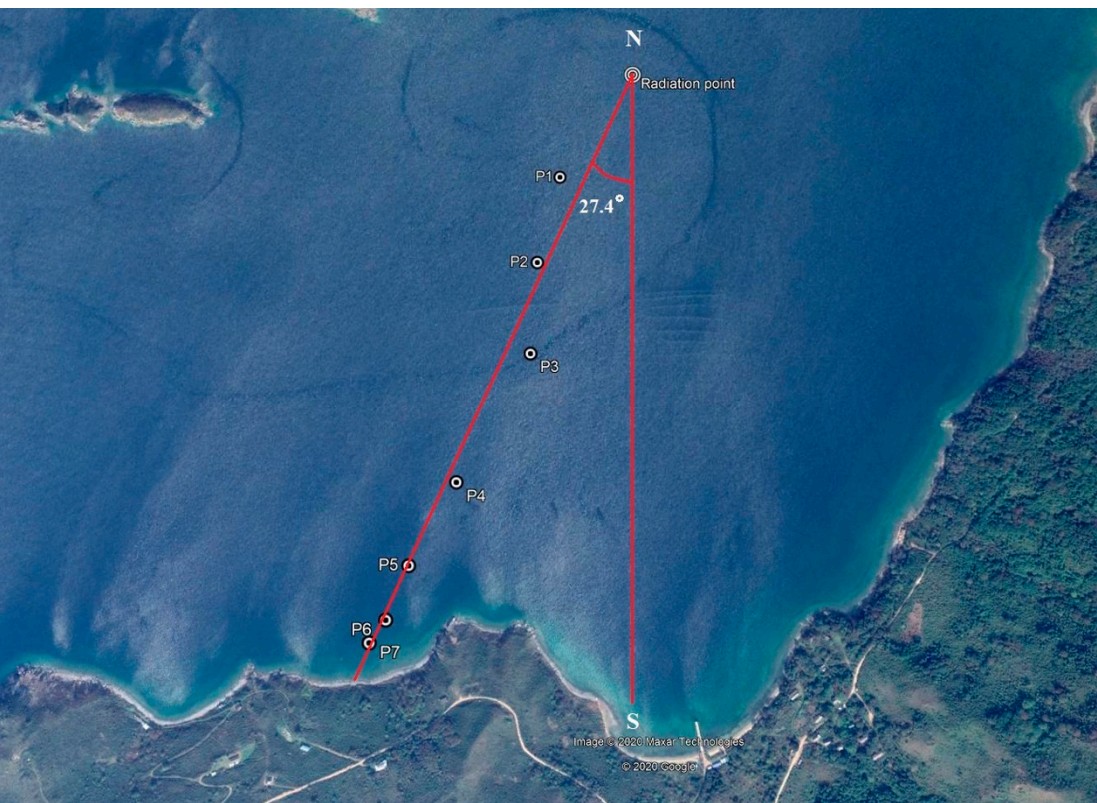

**Figure 1.** Map scheme of the experiment. The labelled "radiation point" is the location of the vessel with the transmitter. Points P1–P7 show the positions of the receiving stations. The angle 27.4° is the angle between the reception route and the "North–South" direction.

The transmitter was a part of a transmitting hydroacoustic system, consisting of a transmitter with an electromagnetic transducer, a suspension frame for the transmitter, a cable-hose with a control pressure gauge, an electric power supply, an electric pump, a control hydrophone, and two calibration accelerometers. The transmitting hydroacoustic system was designed to generate harmonic and phase-manipulated hydroacoustic signals in the frequency band of about 1 Hz in the range of 19–26 Hz.

The amplitude of the volumetric oscillatory displacements of the transmitter reached 0.0123 m$^3$. At the frequency of 20 Hz in unlimited water, this corresponded to a transmitted acoustic power of 1000 W. The configuration of the transmitting hydroacoustic system is shown in Figure 2.

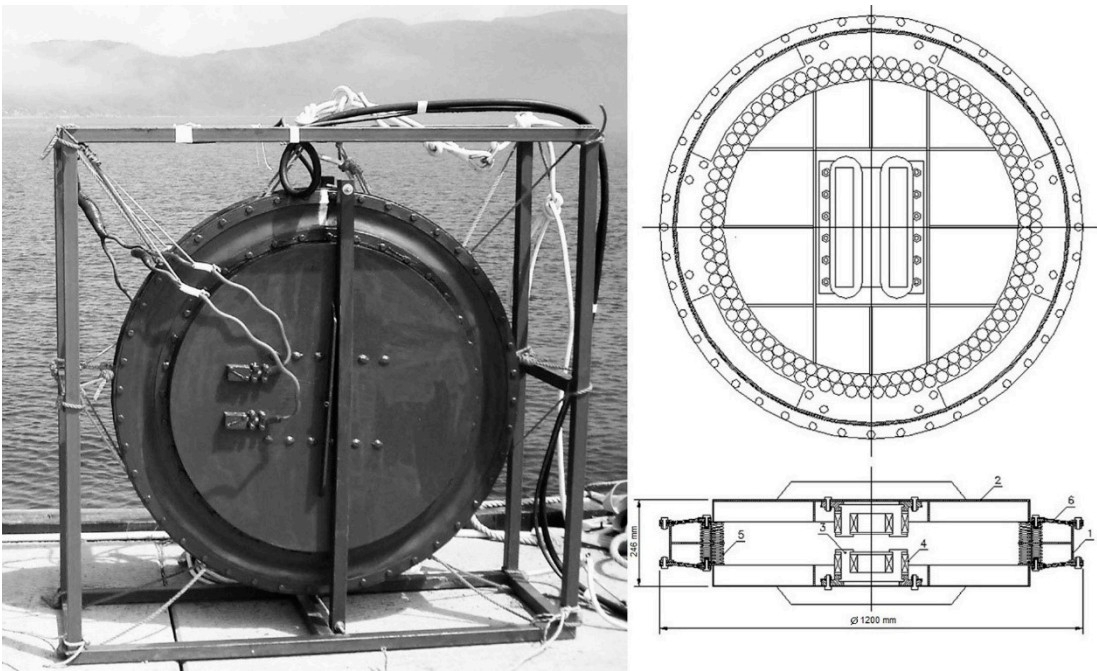

**Figure 2.** Hydroacoustic transmitter of 19–26 Hz.

The transmitter had a mass of 260 kg in air and 40 kg in water. It contains a cylindrical case (1) and a pair of transmitting pistons (2), oscillating in mutually antithetic directions and creating in-phase flows of volumetric oscillatory velocity. The oscillations were induced by the electromagnetic type transducer with U-shaped composite halves of the core (3) and four coils (4). Between the edges of the pistons, a set of 312 cylindrical springs (5) was clamped. Their preliminary compression was achieved due to the decrease in air pressure by 0.5 atm in the transmitter cavity relative to the hydrostatic pressure at the depth of its immersion. To compensate for the hydrostatic pressure during immersion or lifting, a 60 m-long hose with a control pressure gauge and two nipples was used. The gaps between the flanges of the case and the edges of the pistons were sealed with rubber-fabric collars (6). The coils in each pair at half of the core (3) were connected in series. The current was supplied to them by means of elastic conductors put out through seals on the pistons (2). The conductors were commutated directly to the transmitter with terminal-bolt connections with polyisobutylene waterproofing and corresponded to the series connection of the pairs. A battery of series-connected (from 3 to 22 pieces, depending on the required power) acid batteries with a voltage of 12 V and capacity of 90 A*h was used as the primary source of direct current. The power source was a bridge key amplifier made of two half-bridge IGBT modules and equipped with a compensating capacitor battery of 420 μF, a circuit breaker, and a DC ammeter. When operating at stations P3–P7, the low-frequency hydroacoustic transmitter created a pressure of about 7 kPa, and when operating at stations P1–P2, about 5.8 kPa.

The transmitted hydroacoustic signal was recorded with a Bruel & Kjaer 8104 hydrophone at different depths at points P1–P7 of the study ground (see Figure 1); it was immersed from aboard the vessel. The hydrophone was placed on a mobile capsule containing a hydrophone preamplifier, an analog-to-digital converter, and an autonomous recording device. At each reception point, the recording was carried out every 1 m from the surface to the bottom.

The hydrophone was used for the vertical sounding of the hydroacoustic field near the transmitter, on which, subsequently, the energy density created in the water by the transmitter was determined.

On the shore, the transformed seismoacoustic signals were received by a 52.5 m shore laser strainmeter, which was located at Schultz Cape at a point with coordinates N 42°34.798′, E 131°09.400′. The operation process of the laser strainmeter is briefly described in [3]. The part of the 52.5 m laser strainmeter (corner reflector) closest to the water was located at a distance of 120 m from the water edge and at an altitude of 67 m above sea level. The operating arm of the 52.5 m laser strainmeter was oriented at an angle of 180 degrees (1980) to the "North–South" line. All the information from the registration system of the laser strainmeter was fed via cable lines to the laboratory building, where it was entered into a specially created database. Subsequently, the obtained experimental data underwent preliminary and final processing depending on its purpose. The methods of interferometry applied in the laser strainmeter allowed the measuring of the change in the intensity of the interference pattern with high accuracy. Theoretically, using interference methods, it was possible to measure the displacement between the plates of the strainmeter with an accuracy of $10^{-6} \times \lambda_l/2$, where $\lambda_l$ is the wavelength of the applied frequency-stabilized laser. In our case, the measuring accuracy of the displacement between the plates was 0.01 nm. The 52.5 m laser strainmeter used was a frequency-stabilized helium–neon laser that provided a long-term frequency stability of up to the eleventh digit. With a measurement accuracy of 0.01 nm, the sensitivity of the 52.5 m laser strainmeter was $\Delta L/L = 0.01$ nm/52.5 m = 0.19 $\times 10^{-12}$. The amplitude–frequency characteristics of the 52.5 m laser strainmeter had a linear form in the infrasonic range, and in the sound range were the square of sine with doubled amplitude [5]. When registering a longitudinal-type seismoacoustic wave propagating along the axis of the laser strainmeter at a speed of 2300 m/s and taken in accordance with [3], it registered the wave amplitudes at frequencies of 22 and 33 Hz, equal to $2A_0$ and $0.98A_0$, respectively, where $A_0$ is the amplitude of the seismoacoustic wave. This must be taken into account in further calculations, which were carried out using the data from the laser strainmeter.

## 3. Experimental Data Processing

Each device used in the experiment was "tied" to the exact time clock, providing an accuracy of 1 ms. During pre-processing, spectral processing of the obtained synchronous data of the hydrophone and the laser strainmeter was carried out. It was found that in the data of both receiving systems there were oscillations at a frequency of 22 Hz when operating at stations P1–P7. As an example of the stable reception of the transmitted signal by the hydrophone and the laser strainmeter, Figure 3 shows the sections of spectra of simultaneous records by the hydrophone and the laser strainmeter during operation at station 3, in which powerful spectral components at the frequency of 22 Hz are singled out.

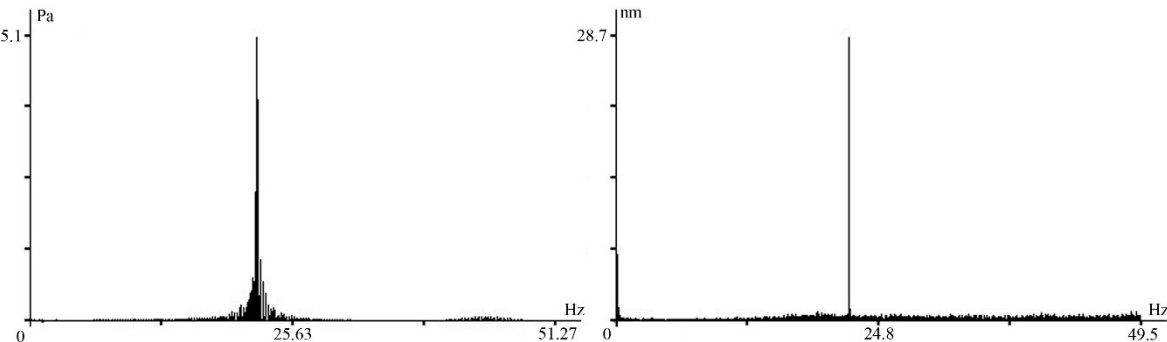

**Figure 3.** Sections of spectra of synchronous data from the hydrophone (**left**) and the laser strainmeter (**right**).

During measurements at each station, the transmitter operated for about 15 min in a continuous mode of transmitting harmonic signals at a frequency of 22 Hz. The measurements by the hydrophone were carried out every 1 m from the bottom to the surface. At each horizon, the hydrophone recorded a signal for 1 min. From the obtained hydrophone data, the amplitude of the received signal was determined at a frequency of 22 Hz. Based on this data, a curve was constructed that describes the

level of the received hydroacoustic signal, starting from the surface to the bottom. For clarity, Figure 4 present the curves of the obtained experimental data (corresponding model curves are also given, which will be discussed in "model calculations").

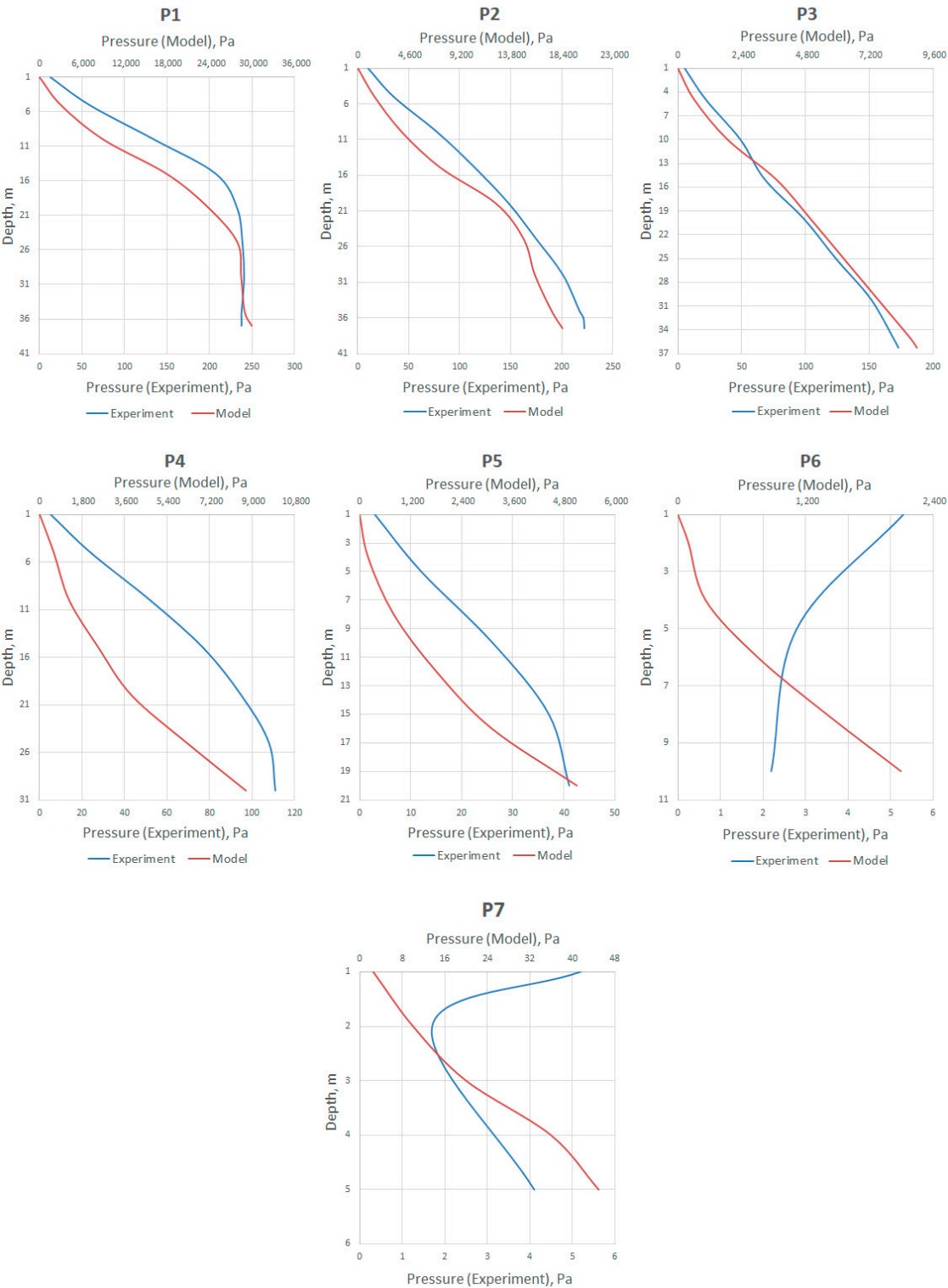

**Figure 4.** Vertical profiles of the distributions of hydroacoustic pressure at a frequency of 22 Hz at stations P1–P7.

For better visualization, the experimental curves presented in Figure 4 were grouped into one figure, which is presented in Figure 5. As we can see in Figure 5, the pressure wave at point P1 has a pronounced section of increased amplitude at a depth of 18 m, associated with the position of the emitter's acoustic axis at the same depth. However, when passing through the points P2–P4, the curves become flatter, which can be explained by energy loss during interaction with the bottom. When passing the critical depth, a sharp drop in the pressure amplitude is seen at the measuring point P5, after which the pressure wave at points P6 and P7 abruptly changes its profile to the opposite.

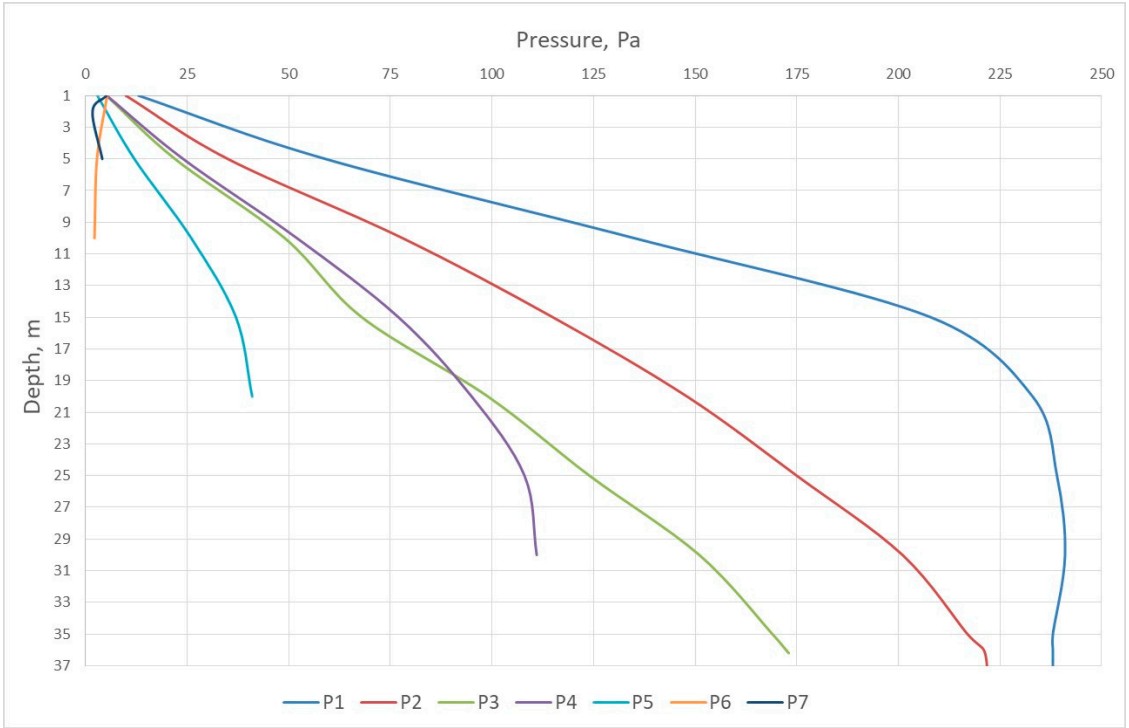

**Figure 5.** Vertical profile of the hydroacoustic pressure distribution at a frequency of 22 Hz at stations P1–P7.

Then, for each experimental curve, a polynomial equation of the most suitable degree was constructed. The experimental data of the laser strainmeter was recorded on the working computer with a sampling frequency of 1000 Hz. When operating at each transmission station, the recording of the laser strainmeter was divided into successive sections of 65536 points which subsequently underwent spectral processing. With such processing, 12–13 sections of the laser strainmeter records were singled out during the operation at each transmission station and, in accordance with this, 12–13 signal amplitudes were obtained at a frequency of 22 Hz, from which the average amplitude was subsequently determined. The results described in this paragraph are shown in Table 1.

In the calculations to be presented below, the following data were used: $\rho_w$ = 1000 kg/m$^3$ (density of water), $c_w$ = 1500 m/s (sound velocity in water), $\rho_b$ = 2100 kg/m$^3$ (rock density of the upper layer of the earth crust), $c_b$ = 2300 m/s (Rayleigh-type wave velocity at the "earth crust–air" boundary [3]), f = 22 Hz (frequency), $\theta$ = 9.4° (angle between the axis of the laser strainmeter and the "transmitter stations P1–P7" line), $\omega$ = 2$\pi$f, R = 1780 m (distance from the point of transmission to the nearest point of the laser strainmeter).

**Table 1.** Experimental data.

| № | Sea Depth, m. | Distance from Transmitter, m. | Polynomial Equation | Wave Amplitude at the Frequency of 22 Hz, Singled out from the Laser Strainmeter Record, nm |
|---|---|---|---|---|
| 1 | 37 | 335 | $S_{37}(z) = (3.2835124*10^{-7})*z^7 - (4.5340116*10^{-5})*z^6 + 0.0024484*z^5 - 0.0643262*z^4 + 0.8225148*z^3 - 4.5942522*z^2 + 21.798353*z - 4.9269574$ | 14.6 |
| 2 | 37.5 | 502 | $S_{37.5}(z) = (8.8507925*10^{-5})*z^4 - 0.0085366*z^3 + 0.1861208*z^2 + 6.2774712*z + 2.4975244$ | 16.4 |
| 3 | 36.2 | 674 | $S_{36.2}(z) = -(1.1161644*10^{-4})*z^4 + 0.0063251*z^3 - 0.0895405*z^2 + 5.0292998*z - 0.206082$ | 28.7 |
| 4 | 30 | 970 | $S_{30}(z) = (1.2446708*10^{-4})*z^4 - 0.0112489*z^3 + 0.2185657*z^2 + 3.8794756*z + 0.9499853$ | 26.7 |
| 5 | 20 | 1148 | $S_{20}(x) = (1.641604*10^{-4})*z^4 - 0.0135414*z^3 + 0.2436404*z^2 + 1.2073308*z + 1.462406$ | 28.1 |
| 6 | 10 | 1252 | $S_{10}(z) = 0.0561111*z^2 - 0.9616667*z + 6.2055556$ | 24.4 |
| 7 | 5 | 1300 | $S_5(z) = 1.075*z^2 - 6.725*z + 10.85$ | 23.3 |

The energy density of the hydroacoustic waves propagating along the shelf were calculated by utilizing the polynomial equations given in Table 1 for all the receiving stations, according to the first formula.

$$E_r = \int_0^h \frac{(S_h(z))^2}{2\rho_w c_w^2} dz, \tag{1}$$

where $S_h(z)$ is the polynomial equation of the curve at sea depth $h$.

In accordance with [3,6], we assume that the signal transmitted by the low-frequency hydroacoustic transmitter reaches the laser strainmeter in the form of surface waves. In this case, the main contribution to the displacement of the earth crust was made by the damped Rayleigh-type surface waves. The amplitude of these waves decays exponentially with the depth of the earth crust. Therefore, the elastic energy density of the Rayleigh-type surface waves can be calculated using the following Equation (2).

$$E_r = \int_0^{\lambda_b} \frac{\rho_b \omega^2 A_0^2 \exp(-4\pi z / \lambda_b)}{2\cos(\theta)^2} dz, \tag{2}$$

where $A_0$ is the displacement amplitude at the frequency of the transmitted signal, equal to half of the wave amplitude at the frequency of 22 Hz, singled out from the laser strainmeter record. $\lambda_b$ is the Rayleigh-type wavelength at the "air–earth crust" boundary and is equal to 104.5 m. The calculation results are shown in Table 2. Table 2 also shows the data for the share of the hydroacoustic energy density at each receiving station that was calculated from the energy density of the transmitter, taking into account the cylindrical divergence of the transmitted signal and the share of the density of the hydroacoustic energy transformed into the density of seismic acoustic energy at each receiving station, taking into account the cylindrical divergence of the Rayleigh-type waves.

**Table 2.** The calculated data.

| Distance from Transmitter, m | Sea Depth, m | Energy Density of the Transmitter, Joul/m³ | Hydroacoustic Energy Density, Joul/m³ | Energy Density of the Transmitter/ Hydroacoustic Energy Density, % | Density of Seismic Acoustic Energy, Joul/m³ | Density of Seismic Acoustic Energy/ Hydroacoustic Energy Density, % |
|---|---|---|---|---|---|---|
| 335 | 37 | 0.124 | $3.1 \times 10^{-4}$ | 83.8 | $0.91 \times 10^{-8}$ | 4.2 |
| 502 | 37.5 | 0.124 | $1.7 \times 10^{-4}$ | 68.8 | $1.15 \times 10^{-8}$ | 8.8 |
| 674 | 36.2 | 0.176 | $8.3 \times 10^{-5}$ | 31.8 | $3.51 \times 10^{-8}$ | 46.8 |
| 970 | 30 | 0.176 | $4.0 \times 10^{-5}$ | 22.1 | $3.04 \times 10^{-8}$ | 61.5 |
| 1148 | 20 | 0.176 | $3.3 \times 10^{-6}$ | 2.2 | $3.37 \times 10^{-8}$ | 100 |
| 1252 | 10 | 0.176 | $1.9 \times 10^{-8}$ | 1.4 | $2.54 \times 10^{-8}$ | 100 |
| 1300 | 5 | 0.176 | $4.3 \times 10^{-9}$ | 0.3 | $2.32 \times 10^{-8}$ | 100 |

## 4. Model Calculations

With the purpose of studying the spatial distribution of the hydroacoustic energy, model calculations were performed. The simulation tool is the spectral element method (SEM), which is a high-order finite element method developed for the local and global scales of seismic wave propagation [7]. We simulated the propagation of an acoustic signal at the shelf of decreasing depth using the SEM software package with the SPECFEM2D open-source code [8,9].

SPECFEM2D integrates a simplified form of the wave Equation (3) using a high degree Lagrange interpolation polynomial. In a spatially inhomogeneous area of liquid, the wave equation for the pressure $P(x,t)$ has the form

$$\frac{1}{k}\ddot{P} = \nabla \cdot \left(\frac{\nabla P}{\rho}\right),$$

(3)

where $k(x)$ is the adiabatic volumetric modulus of liquid elasticity.

In linearly elastic bodies, the strain tensor $\varepsilon(x,t)$ is calculated by the displacement vector u as

$$\varepsilon = \frac{1}{2}\left(\nabla \vec{u} + (\nabla \vec{u})^T\right).$$

(4)

The stress tensor can be expressed through the strain tensor $\vec{\sigma}(\vec{x},t)$ according to Hooke's law.

$$\vec{\sigma} = \vec{c} : \varepsilon,$$

(5)

where the colon denotes the double tensor reduction operation. The elastic properties of the medium are described by a fourth-order elastic tensor.

SPECFEM2D allows the users to perform 2D and 2.5D (axisymmetric) simulations of the propagation of acoustic, elastic, visco-elastic, and poro-elastic acoustic waves. In addition, SEM combines the flexibility of the finite element method with the accuracy of the spectral method. Therefore, this software package is appropriate for performing numerical simulations in ocean acoustics with complex media and topography.

The model is shown in Figure 6. The ocean surface is set at z = 0 m. All the physical parameters of the medium are listed in Table 3. The calculation area has a width of 3000 m and a depth of 100 m below the sea surface. All sides, except the top and axis, are covered with perfectly matched layers (PML) [10]. The source of Ricker pressure pulses with a dominant frequency of 22 Hz is located at (rs, zs) = (0, −18 m). The grid of the model consists of 118,385 spectral elements constructed using the open source Gmsh software [11]. Each numerical simulation takes an hour on a PC with the Linux operating system.

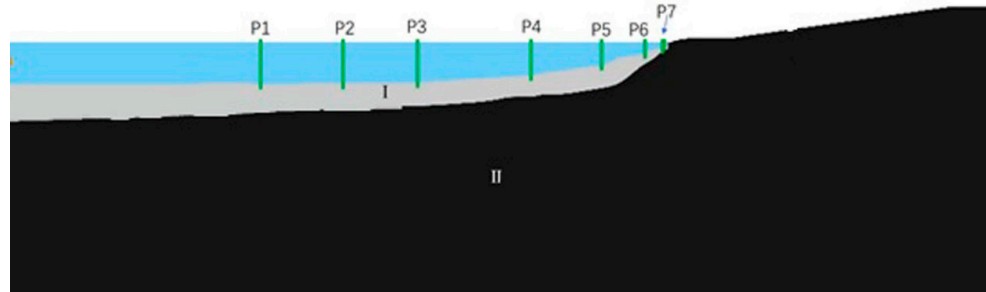

**Figure 6.** Solution domain.

**Table 3.** Medium parameters used for the simulation.

|  | $c_p$ | $c_s$ | $\rho$ |
|---|---|---|---|
| Precipitation | 1600 m/s | 400 m/s | 1.9 g/sm$^3$ |
| Semi-infinite granite space | 5400 m/s | 3300 m/s | 2.79 g/sm$^3$ |

## 5. Analysis of the Results

Based on the data of the model calculations at certain points on the route that coincided with sounding points, vertical profiles of pressure distribution from the surface to the bottom were constructed. Model pressure distribution curves at the transmitter operating frequency of 22 Hz from the surface to the bottom are shown against the experimental curves in Figure 4. Based on these curves, polynomial equations of the most suitable degree were constructed according to which the model energy density at each receiving station was calculated by means of Equation (1) and the model energy density was calculated at each of the receiving stations (for stations P1–P7: 3.715, 1.133, $2.14 \times 10^{-1}$, $1.08 \times 10^{-1}$, $2.18 \times 10^{-2}$, $2.05 \times 10^{-3}$, $6.30 \times 10^{-7}$ J/m$^3$). The graph of their changes is shown in Figure 7. The graph of the experimental data changes is also shown in this figure. All the obtained model calculations provided overestimated results in comparison with the experimental data, which, in our opinion, is due to the fact that the model is very approximate and does not take into account many of the effects of the propagation of hydroacoustic waves in the shallow sea, and also does not take into account the contribution of the damped and undamped Rayleigh-type waves spreading along the "water-bottom" boundary. Nevertheless, at many of the receiving stations, the model curves are similar to the experimental ones and display a relatively reliable distribution of the pressure created by the hydroacoustic transmitter at a frequency of 22 Hz from the surface to the bottom.

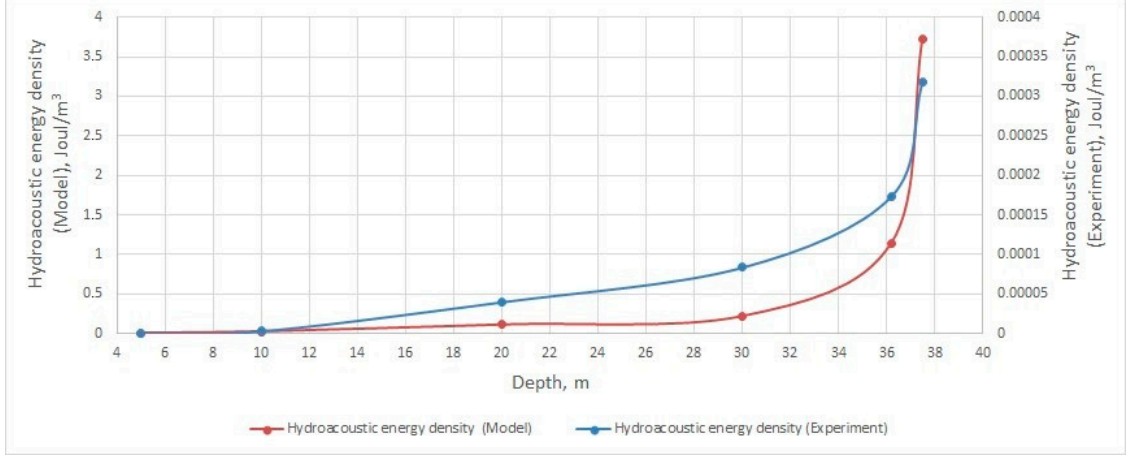

**Figure 7.** Graphs of the hydroacoustic energy density changes in the model calculations and experimental research.

A significant difference in the behavior of the model and the experimental curves was noted at stations 6 and 7 (see Figure 4 P6 and P7, respectively).

Judging by the upper part of the drawing and the slope of the curve to the right and not to the left (as model calculations show), this contribution was made by an undamped Rayleigh-type wave propagating along the "water-bottom" boundary, the value of the generated pressure of which is about 4–5 Pa, which is also consistent with the calculated data given in Table 2 for these stations in the column "Transmitter/Hydroacoustics, %" and well as the conclusions of [12].

Let us analyze in more detail the experimental results given above. In the analysis, we will take into account that a hydroacoustic wave propagates along the wedge-shaped shelf. The loss of energy of the hydroacoustic wave propagating along the wedge-shaped shelf is determined by several factors. If the depth of the sea is several times greater than the wavelength, then the main causes of energy loss are sound absorption by water and its scattering by the sea surface. With decreasing frequency, the influence of the sea bottom begins to prevail in the attenuation of the hydroacoustic wave. This effect manifests itself as an increase in the dissipation of wave energy by the bottom soil. For wavelengths greater than the depths of the sea, the fraction of energy that is absorbed from the water layer into the sea bottom and is converted into elastic waves of the water-bottom interface increases. With an increase in the wavelength, the generation of Rayleigh waves and body waves begin to play a major role in attenuating the energy of the hydroacoustic wave.

The influence of the wedge on the frequency and spatial dependence of the energy of the hydroacoustic wave emitted from the water layer can be estimated as follows. The pressure of the hydroacoustic field with the frequency $\omega = 2\pi f$, generated by the source at a given point with the coordinates $(x, z)$ can be represented as the sum of the normal waves (modes) [13]:

$$P(x,z,t) = \sum_l \left( \frac{2P_0 R}{H} \left( \frac{2\pi}{\xi_l x} \right)^{1/2} \sin(\alpha_l z) \sin(\alpha_l z_1) e^{i(\xi_l x + \frac{\pi}{4})} \right) e^{-i\omega t}, \tag{6}$$

where $H$ is the depth of the sea, $P$ is the pressure amplitude at the distance R from the emitter, and $\xi_l$ and $\alpha_l$ are horizontal and vertical wave numbers of the mode l.

In this case, the group wave velocity can be written as $U_l(f) = c\left(1 - \frac{f_l^z}{f^z}\right)^{1/z}$, where $f_l$ is the critical frequency of the mode number $l$. At frequencies less than critical, the hydroacoustic wave becomes inhomogeneous and the energy completely transfers from hydroacoustic to seismo- acoustic. In case of the shallow sea, when the propagation condition is satisfied only for one mode ($l = 0$), the critical frequency can be found by the formula $f_0 = \frac{c}{4H\sqrt{1-n^2}}$, where $n$ is the refractive index equal to the ratio of the speed of sound in water to the speed of sound in the sea bottom. Based on the formula for critical frequency, we can find the critical depth $H_{cr}$ at which, at a given frequency of the hydroacoustic wave, all the energy will go to the bottom.

Figure 8 shows graphs of dependence for the seismoacoustic and hydroacoustic wave energy densities on the depth of the points where the measurements were taken. The critical depth of the wave with a frequency of 22 Hz was 17.8 m. As we can see from the graphs, a significant part of the hydroacoustic wave energy density goes into the seismoacoustic wave energy density down to a depth of 36 m, which is approximately half the length of a wave with a frequency of 22 Hz (34 m). After the depth equal to that of the critical (17.8 m), it almost completely transforms into the energy density of the seismoacoustic wave.

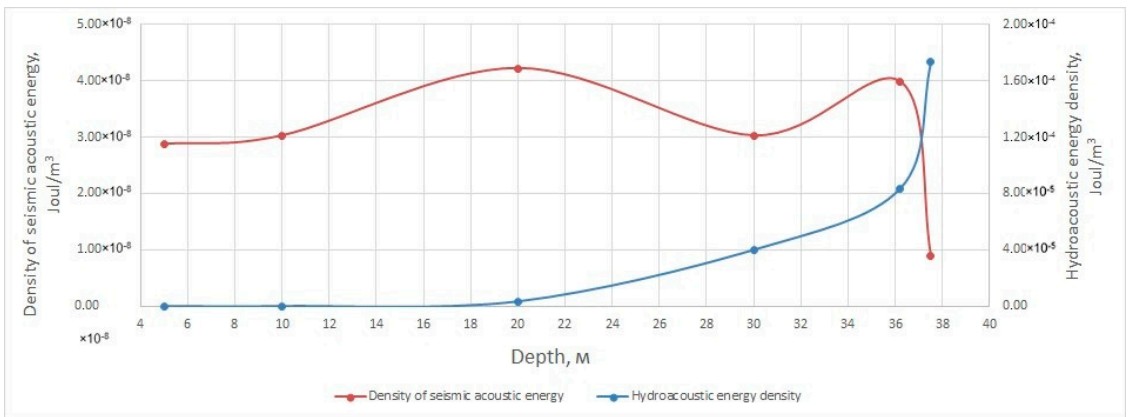

**Figure 8.** Graphs of the dependence of the seismoacoustic and hydroacoustic wave energy densities on the depth of the points where the measurements were taken.

In the case of the experiment with a hydroacoustic emitter at a frequency of 33 Hz [3] with the same model of the seabed, the critical depth should be 11.8 m. The emitting stations described in [3] were located at significant distances from each other, which did not allow us to experimentally determine the critical depth for the frequency of 33 Hz.

Figure 9 shows graphs of the energy density of the seismoacoustic wave recorded by the laser strainmeter and the difference in the energy density of the experiment and the cylindrical divergence in percent. As you can see, the graphs are almost identical, which can tell us that almost all the energy losses are along the path of the hydroacoustic wave, from the first measurement point to the last transfer into the elastic oscillations of the earth crust.

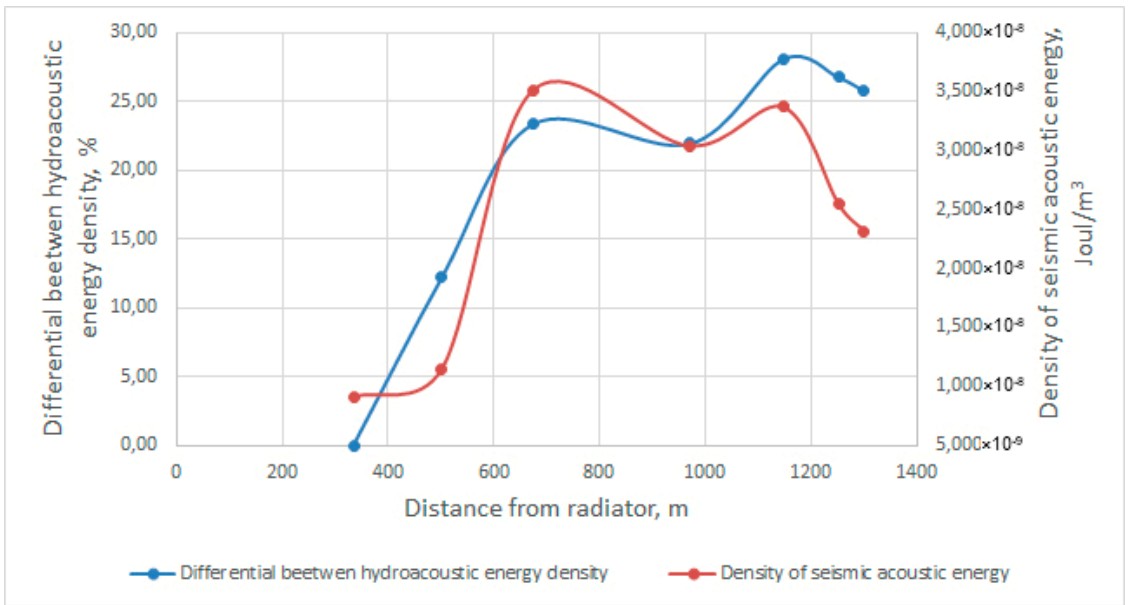

**Figure 9.** Graphs of the energy density of the seismoacoustic wave recorded by the laser strainmeter, and the difference in the energy density of the experiment and the cylindrical divergence in percent.

Further, let us analyze the calculated data shown in Table 2. We perform the analysis accounting for the fact that the hydroacoustic energy is distributed along the shelf according to the cylindrical law and its absorption at such small distances is negligible. Suppose that the energy has come in full to the first measurement point, i.e., the losses amounted to 0%, both in the case of the experiment and in the case of cylindrical divergence. Then, we can calculate in terms of percent how much energy went into the bottom in the form of body waves and did not reach the laser strainmeter. As we can

see from the fifth column, at the distance of 335 m from the emission site, 83.8% of the energy given out by the emitter transferred into hydroacoustic energy. That is, about 16% went into the bottom in the form of body waves. If we compare this with the results of [3], obtained in the experiment with a low-frequency hydroacoustic emitter at 33 Hz, we can state that when the hydroacoustic emitter was operating at 22 Hz, much less energy went into the body waves than when the hydroacoustic emitter was operating at 33 Hz (78% [3]).

From the theory of waveguide sound propagation, we can estimate the number of normal wave modes that emerge when a harmonic signal is emitted with frequency f for the waveguide depth h using the expression $l < kh/\pi$, where $k$ is the wave number and $l$ is the number of mode of the normal wave. In this case, there is a condition for locking the waveguide $kh < \pi$ (the waveguide depth is less than half the wavelength of the acoustic signal) under which only the zero mode can propagate. Thus, when the emitter is operating at 33 Hz at a depth of 30 m, two modes of normal waves should propagate—zero and first—while, when emitting 22 Hz at a depth of 32 m, only zero mode should propagate. We should note that with the effect of locking the waveguide and the propagation of several modes of normal waves, taking into account the principle of superposition, the pressure distribution across the waveguide remains the same. Moreover, high-order modes are inhomogeneous and quickly decay or transfer to other forms of energy, in contrast to the zero mode, which can propagate over long distances. Given all of the above, we can explain the large amount of hydroacoustic energy remaining in the water at 22 Hz emission by the effect of locking the waveguide, when only the zero mode appeared with concentrations of almost all the emitted energy in it. Whereas, in the experiment with the 33 Hz emitter, this effect did not occur, two modes were excited here. Subsequently, the energy from the first mode at a short distance from the emitter transferred into seismoacoustic energy in the form of body waves. The zero mode spread further along the water layer of the shelf with an energy equal to 22% of the energy generated by the emitter.

As a result of the performed model calculations that were presented in the previous chapter of this article, a picture of the spatial distribution of hydroacoustic energy from the point of transmission was obtained and is shown in Figure 10.

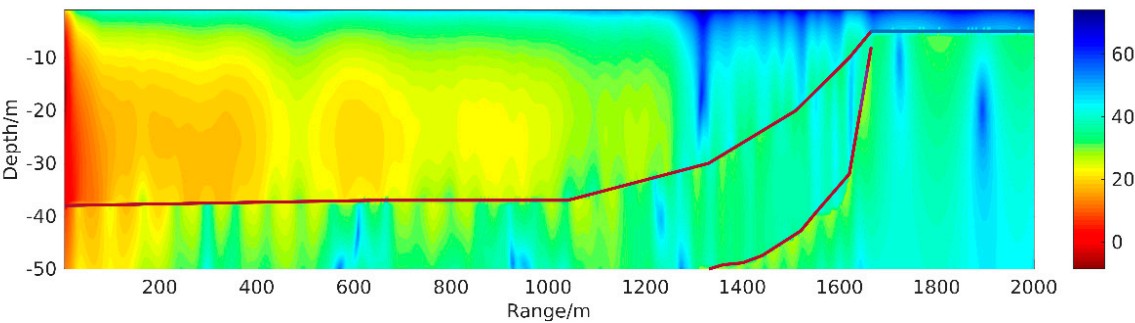

**Figure 10.** Spatial distribution of hydroacoustic energy from the transmitter to the coast along the receiving stations.

Figure 10 shows that the main portion of the energy generated by the hydroacoustic emitter is concentrated in water near it. Only a minor part of it goes to the bottom. While moving along the shelf of decreasing depth, the energy level decreases, which is associated with both cylindrical divergence and the transformation of hydroacoustic energy into seismoacoustic. In this case, the area of the highest concentration of hydroacoustic energy in the initial stage of its propagation from the emitter is at depths of 21–35 m. Moreover, careful examination of the figure shows that the horizontal energy level does not decrease monotonously, but changes "quasi-periodically", the scale of which is connected with the length of the hydroacoustic wave and is equal to 68 m. From the same figure, we can see that at depths slightly over 30 m, there is an abrupt decrease in hydroacoustic energy in the water, and it further drops at depths of about 16–17 m. The presence of energy in the water at shallower depths can only be explained by the undamped waves of the Rayleigh type leaking into the water.

## 6. Conclusions

As a result of the experimental and model studies, the general patterns of the low-frequency hydroacoustic waves' propagation at the shelf of decreasing depth and their transformation into surface-type seismoacoustic waves at the water-bottom interface were identified. It was established that at sea depths of more than half of the hydroacoustic wave by 9%–10%, about 4%–7% of the energy of the hydroacoustic waves was transformed into seismoacoustic energy of the Rayleigh type. With decreasing depth, the percentage of the transformed energy abruptly increased.

The obtained experimental results allowed us to determine the critical depth at which the shelf "locks" the passage of hydroacoustic energy in water at a frequency of 22 Hz. This depth coincides very well with theoretical calculations and is equal to 17.8 m. We can expect that with a decrease in the frequency of the emitted hydroacoustic signal, this effect will manifest itself at greater depths. Thus, for example, when a hydroacoustic signal is emitted at a frequency of 1 Hz at the shelf of decreasing depth and with similar elastic parameters of the seabed, this signal will not be in the water and will start from depths of about 390 m.

**Author Contributions:** Conceptualization G.I.D., data curation S.G.D., V.A.C., S.S.B. and S.V.Y., format analysis G.I.D., S.S.B., S.G.D. and V.A.C., funding acquisition S.P., investigation G.I.D. and S.S.B., methodology G.I.D., project administration S.G.D., software Y.S., X.W., Y.D. and S.S.B., supervision S.G.D. and S.P., visualization Y.S., X.W., Y.D., S.S.B., V.A.C. and S.G.D., writing original draft G.I.D., S.S.B. and S.P., writing review and editing S.G.D. All authors have read and agreed to the published version of the manuscript.

**Funding:** This work was carried out with partial financial support of the topic AAAA-A20-120021990003-3 "Research of fundamental foundations of the origin, development, transformation and interaction of hydroacoustic, hydrophysical and geophysical fields of the World Ocean".

**Conflicts of Interest:** The authors declare no conflict of interest.

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
