# Peer review of "Study of Low-Frequency Hydroacoustic Waves’ Behavior at the Shelf of Decreasing Depth"

_applsci, doi:10.3390/app10093183_

Round 1

Reviewer 1 Report

Congratulations to the authors. They have much improved their article in the  revised version. According to my opinion the paper can now be published. I would prefer to incorporate some language corrections of mine at some points in the attached text in the form of comments. I don't feel that I have to see the paper again.

Author Response

Thank you for your work

Reviewer 2 Report

The article describes an interesting study on the impact of sea depth on the conversion of hydroacoustic to seismic waves. Experiments were carried out in a real environment, involving significant resources. The results allow to estimate at what depth of the sea a significant part of the sound energy of the wave propagating in water will be changed into the energy of seismic acoustic waves. This depth depends on the frequency of the sound. The obtained results allow to predict the manner of propagation of complex sounds.
Here are some comments about the text. Out of curiosity, I entered in Google Earth the coordinates given in the article. They do not exactly coincide with the map in the article, e.g. point N42°35.5817', E131°09.9667' is not on the shore, but at sea (near the transmitter). In figure 1, there is no point "Unnamed label" (as written in line 55), only "Radiation point".
The measurement results presented require more accurate descriptions and corrections in some places. For example, Fig. 4 shows the theoretical and observed dependence of hydroacoustic pressure on depth for measuring points at different distances from the sound source. In particular, in the last figure (P7) the experimentally determined pressure exceeds 40 Pa for a depth of 1 meter. But in collective Fig. 5, the pressure value for P7 and this depth is 10 times smaller. This matches with the polynomial in Table 1s. Thus, the axis in figure 4 / P7 is probably incorrectly described. I think other descriptions require checking. The polynomials in Table 1 needlessly have so many significant digits in the coefficients. In Table 2, I was unable to get the value from the last column by doing the division given in the header. In Figure 7, it was not clear to me where the difference by a factor of 10,000 between the theoretical and measured value of hydroacoustic pressure came from.

Author Response

Thank you for your work. Responses to comments in the file.

This manuscript is a resubmission of an earlier submission. The following is a list of the peer review reports and author responses from that submission.

Round 1

Reviewer 1 Report

In this reviewer's opinion, the paper could be published bau after a moderate revision. See comments in the attached file.

An important remark is that although the introduction refers to a previous similar work of the authors in order to justify the present one, in the Analysis of results and conclusions section almost no reference to that previous work is done! The present results should be compared (qualitatively) with the previous ones.

Reviewer 2 Report

The authors present experimental results performed at the Sea of Japan about transmission and transformation of hydroacoustic signals.

The topic is certainly interesting, but I am not very enthusiastic about this paper. The quality of the presentation is rather low. The discussion of the results is superficial, and the main findings appear to be very similar to those reported in previous papers by the same authors. Therefore, I will not recommend publication.